# Improving the Quality and Safety of Fresh Camel Meat Contaminated with *Campylobacter jejuni* Using Citrox, Chitosan, and Vacuum Packaging to Extend Shelf Life

**DOI:** 10.3390/ani11041152

**Published:** 2021-04-17

**Authors:** Hany M. Yehia, Abdulrahman H. Al-Masoud, Manal F. Elkhadragy, Shereen M. Korany, Hend M. S. Nada, Najla A. Albaridi, Abdulhakeem A. Alzahrani, Mosffer M. AL-Dagal

**Affiliations:** 1Department of Food Science and Nutrition, College of Food and Agriculture Sciences, King Saud University, Riyadh 11451, Saudi Arabia; amasood@ksu.edu.sa (A.H.A.-M.); aabdulhakeem@ksu.edu.sa (A.A.A.); maldagal@ksu.edu.sa (M.M.A.-D.); 2Department of Food Science and Nutrition, Faculty of Home Economics, Helwan University, Cairo 11221, Egypt; 3Department of Biology, Faculty of Science, Princess Nourah Bint Abdulrahman University, Riyadh 11671, Saudi Arabia; mfelkhadragy@pnu.edu.sa (M.F.E.); shirienmagdy@yahoo.com (S.M.K.); 4Department of Botany and Microbiology, Faculty of Science, Helwan University, Cairo 11421, Egypt; 5Department of Microbiology, Faculty of Veterinary Medicine, Zagazig University, Zagazig 44519, Egypt; Hend.saeed@hotmail.com; 6Nutrition and Food Science, Department of Physical Sport Science, Princess Nourah bint Abdulrahman University, Riyadh 11671, Saudi Arabia; naalbaridi@pnu.edu.sa

**Keywords:** camel meat, *C. jejuni*, citrox, chitosan, total volatile base nitrogen (TVB-N), sensory evaluation

## Abstract

**Simple Summary:**

This study aimed to investigate the influence of using 1% or 2% Citrox alone or in combination with 1% chitosan on the survival of *Campylobacter jejuni* in camel meat slices vacuum-packed and stored at 4 or 10 °C for 30 days. The shelf life of camel meat was 30 days longer using 1% or 2% Citrox in combination with 1% chitosan than when using Citrox alone. The reductions ranged from 4.0 to 3.5 logarithmic cycles during the storage period at both 4 and 10 °C. The quality of camel meat treated with Citrox plus chitosan was also better than that of the control meat and of meat treated with 0.85% NaCl.

**Abstract:**

Camel meat is one of the most consumed meats in Arab countries. The use of natural antimicrobial agents to extend the shelf life of fresh camel meat, control *Campylobacter jejuni* contamination, and preserve meat quality is preferred. In this study, we determined the antimicrobial effects of using 1% or 2% Citrox alone or in combination with 1% chitosan on the survival of *C. jejuni* in vitro and on camel meat samples during storage at 4 or 10 °C for 30 days in vacuum packaging. We determined the total viable count (TVC (cfu/g)), total volatile base nitrogen (TVB-N) content, and pH of the treated camel meat samples every three days during storage. The shelf lives of camel meat samples treated with 2% Citrox alone or in combination with 1% chitosan were longer than those of camel meat samples treated with 1% Citrox alone or in combination with 1% chitosan at both the 4 and 10 °C storage temperatures, with TVCs of <100 cfu/g after the first ten days and six days of storage at 4 and 10 °C, respectively. The addition of Citrox (1% and 2%) and 1% chitosan to camel meat samples and the application of vacuum storage were more effective than using Citrox (1% and 2%) alone and led to a reduction in *C. jejuni* in approximately 4.0 and 3.5 log cycles at 4 and 10 °C, respectively. The experimental results demonstrated that using a Citrox-chitosan combination improved the quality of camel meat and enhanced the long-term preservation of fresh meat for up to or more than 30 days at 4 °C.

## 1. Introduction

One of the most common causes of bacterial gastroenteritis consists of *Campylobacter* species, which are distributed globally. The worldwide rate of *Campylobacter* infection has increased in the last decade [1,2], and such infections have been confirmed to pose severe public health risks [3]. *Campylobacter* infection is a zoonosis that causes foodborne illness [4]. The consumption of undercooked red meat or poultry [5] is considered a risk factor [6] for *Campylobacter* infection. Foods containing meat or animal products can be contaminated with *Campylobacter* during slaughter and carcass dressing [7,8,9]. Meat and meat products, such as burgers and sausages, contain high protein levels and are regular components of human diets. Therefore, their hygiene is essential to public health, as the consumption of poor-quality meats can cause infections [10]. Foodborne bacteria, such as *Campylobacter* species, are typical foodborne pathogens and are responsible for outbreaks [11,12].

Camel meat contains high-quality protein and has low fat and cholesterol contents and higher polyunsaturated fatty acid contents than meat from other animals [13]. In the last decade, camel meat has received increasing attention because it has caused unrecognized foodborne diseases in developing countries [14]. Zoonotic microbes, such as *Enterococcus* sp., *Staphylococcus aureus*, and *Campylobacter* spp., and many other foodborne pathogens contaminate meat. The gastrointestinal tract of food-producing animals can harbor many of these pathogens, which can contaminate meat during processing at an abattoir, resulting in subsequent human illnesses [15]. The Kingdom of Saudi Arabia is projected to be the largest consumer of camel meat in the Arabian Gulf region, and the demand for camel meat is expected to grow as the population, number of visitors, and consumer preference for fresh camel meat increase.

The use of chemical additives in foods has risen, while the safety of these chemicals has been improved. However, worldwide consumer demand for the use of natural products to preserve foods has increased. Recently, companies have been manufacturing Citrox to meet this demand. Due to its plant origins, including citrus extract, citric acid, and polyphenols, it complies with the requirements of European Regulation 2092/91 and EC Directive 89/107/EEC. Citrox interacts with organic matter and impacts microorganisms in various ways, including the deterioration of bacterial films, the breaking down of biofilms, and the reducing of foodborne pathogens, which extend the shelf life of meat. Citrox can be applied directly to food as an additive and conforms to BS EN 1276 (European suspension test).

Citrox includes a range of bioflavonoids derived from citrus fruits. Bioflavonoid and organic acid compounds are effective against viruses, bacteria, molds, and yeasts and exhibit synergistic activity. Citrox can be described as follows: It is composed of natural organic acids and has broad-spectrum activity against microorganisms (Gram-negative and Gram-positive bacteria), molds, yeasts, and viruses. It is safe and exerts its effects while being volatile, nonmutagenic, nontoxic, noncorrosive, nontainting, and noncarcinogenic. It has the ability to break down bacterial biofilms, appears active in the presence of organic matter and debris, and can destroy cell walls.

Chitosan is an aminopolysaccharide that contains a positive electrical charge and bonds to negatively charged molecules. Chitosan and its derivatives are of interest because they are nontoxic antimicrobial and biodegradable biopolymers with many functions [16,17,18]. Chitosan is soluble in various acids, such as hydrochloric, nitric, acetic, phosphoric, and perchloric acids [19,20,21].

The solubility of chitosan in aqueous acidic solutions with a pH lower than 6.5 can be altered and depends on the depolymerization and chemical modification of the primary and secondary hydroxyl groups [22]. Chitosan has antimicrobial activities against Gram-positive and Gram-negative bacteria [23], so it is considered a promising agent for extending the shelf life of food, preserving fresh food, and maintaining food safety.

Chitosan can be used alone or in combination with organic matter to increase its beneficial effects. Roller et al. (2002) [24] combined chitosan with glutamate, sulfite, and carnocin and added the compounds to pork sausages for preservation.

The biomarker which is used as a protein and amine degrader in meat is considered to be total volatile basic nitrogen (TVB-N). Therefore, measurements of total volatile bases nitrogen (TVB-N) can help in the diagnosis of meat contamination. TVB-N values are raised in meat during storage and aligned with other biomarkers of spoilage (i.e., duration time of storage, temperature and packaging conditions, etc.). Also, the pH of meat is considered as the main factor in the determination of fresh meat since it can measure the rate of oxidation of myoglobin and lipids, and therefore of the meat itself.

Sensory analysis is also regarded as a useful tool for the qualitative evaluation of foods; an increase in the level of chemical compounds, total viable count (TVC), or physical alteration of the food matrix results in changes in sensory attributes.

To our knowledge, no studies have been conducted using Citrox alone or in combination with chitosan for the treatment of camel meat to evaluate the effects of Citrox on the survival of *C. jejuni* and the ability to extend the shelf life and quality of camel meat. Thus, we used 1% and 2% Citrox alone or combined with 1% chitosan to treat vacuum-packed camel meat slices stored at 4 or 10 °C for 30 days, and we determined the total viable count (TVC) of *C. jejuni* and the total basic nitrogen (TVB-N) and physicochemical and sensory parameters of the meat.

## 2. Materials and Methods

### 2.1. Camel Meat Samples

Camel meat samples were obtained from markets distributed in Riyadh, Saudi Arabia. The samples were transported to the laboratory of Food Microbiology, College of Food and Agriculture Sciences, King Saud University, under cooled conditions and incubated in a refrigerator at 2 °C for the experiments.

### 2.2. Campylobacter jejuni

An inoculum of *C. jejuni* ATCC 33291 (~10^4^ colony-forming units (cfu/mL)) was used. Two types of media were used to activate the strain, *Campylobacter* blood-free selective medium (modified CCDA-Preston) (Oxoid CM 0739, Basingstoke, UK) supplemented with CCDA Selective Supplement (SR 0155) and Bolton Selective Enrichment Broth (Oxoid CM0983, Basingstoke, UK), and then cooled to 45 to 50 °C, and 25 mL of laked horse blood (SR0048) and one vial of Bolton Broth Selective Supplement (Oxoid SR0183 Basingstoke, UK), were aseptically added. The plates were incubated at 42 °C for 24 to 72 h.

### 2.3. Preparation of Citrox

Pure Citrox was prepared by mixing organic acids in 100 mL of distilled water with a citric acid:malic acid:ascorbic acid ratio of 18:18:5 (by weight). The mixture was yellow in color and had a pH of 2.7. Distilled water was used to dilute the Citrox to 1 and 2% solutions, which were then added to a 1% chitosan solution before sterilization at 121 °C for 15 to 20 min. A saline solution of 0.85% NaCl was also prepared in large quantities and sterilized along with the other solutions.

### 2.4. Camel Meat Samples, Inoculation, Treatment, and Packaging

Seven hundred and ninety-two camel meat samples weighing approximately 25 g each were collected. These samples were separated into six groups, A, B, C, D, E, and F, which each contained 198 samples, to determine the TVC, pH, TVB-N, color, and sensory profile.

#### Treatment Groups

In **Group A,** camel meat samples were immersed in an aluminum tray containing 1 L of 0.85% NaCl and left for 2 min. Group A was the negative control.

**Group B** consisted of camel meat-salt-*C. jejuni*. The samples were immersed in 1 L of 0.85% NaCl, left to drain on a sterilized sieve, placed in a *C. jejuni* inoculum (~10^4^ log cfu/mL), and left for 2 min. Group B was the positive control.

In **Groups C** and **D**, camel meat samples were soaked for 2 min in saline solution, placed into a *C. jejuni* inoculum, and then into either 1% or 2% Citrox, and left for a duration suitable to drain the solutions.

**Group E** consisted of camel meat-salt-*C. jejuni*-1% Citrox and 1% chitosan, as described above.

**Group F** consisted of camel meat-salt-*C. jejuni*-2% Citrox and 1% chitosan, as mentioned above.

All the samples were packed in transparent polyethylene containers (16 × 22 cm), vacuum-sealed (PlusVac 20, KOMET Plochingen, Germany), and stored under cooled conditions in incubators at 4 or 10 °C for 30 days. Samples were analyzed every three days for the determination of the TVC, pH, TVB-N, and sensory profile.

### 2.5. In Vitro Activity of Citrox and Chitosan Towards C. jejuni

This study evaluated the effects of using 1% or 2% Citrox alone or in combination with 1% chitosan on *C. jejuni* in *Campylobacter* blood-free selective medium (modified CCDA-Preston) (Oxoid CM 0739, Basingstoke, UK) supplemented with CCDA Selective Supplement (Oxoid SR 0155, Basingstoke, UK) at 42 °C for 24 to 48 h.

One hundred microliters of *C. jejuni* ATCC 33291 culture (10^6^ cfu/mL) was spread using a cotton swab on the surface of *Campylobacter* blood-free selective medium (modified CCDA-Preston) (Oxoid CM 0739, Basingstoke, UK) supplemented with CCDA Selective Supplement (Oxoid SR 0155, Basingstoke, UK). Using the agar diffusion method, a hole with a diameter of 8 mm was punched aseptically using a sterile cork borer, and 100 µL of 1% or 2% Citrox solution was introduced into each well. To compare its effects on *C. jejuni*, 1% chitosan was mixed with the same volume of 1% or 2% Citrox. The plates were then incubated at 42 °C for 48 to 72 h under a microaerophilic atmosphere in the presence of CO_2_. The diameter (mm) of the zone of inhibition was determined.

### 2.6. Microbiological Analysis

Ten grams of a camel meat sample was transferred to 90 mL of saline solution (0.85% NaCl), placed in a stomacher bag (Seward Ltd., London, UK), and homogenized for 2 min at room temperature. *C. jejuni* cells were counted by using the pour-plate method with serial dilution (1/10–1/10^3^). One milliliter of each sample was mixed with *Campylobacter* blood-free selective medium agar (modified CCDA-Preston, Oxoid CM0739, Basingstoke, UK) supplemented with CCDA Selective Supplement (SR0155, Oxoid CM375, Basingstoke, UK) and incubated at 42 °C for 48 to 72 h under a microaerophilic atmosphere in the presence of CO_2_.

### 2.7. Chemical Analysis

#### 2.7.1. Total Volatile Base Nitrogen (TVB-N)

TVB-N was determined according to the method from the AOAC (1990) [25]. Five grams of a camel meat sample was added to 300 mL of distilled water in a heating flask, and then 2 g of magnesium oxide and some granules of antifoaming pellets were added. The receiving flasks contained 25 mL of boric acid (2%) and a few drops of Tashiro’s indicator (consisting of 1.25 g methyl red and 0.32 g methylene blue dissolved in 1 L of 90% ethanol). The flasks used for the heating and receiving were connected to an evaporator, and the water bath temperature was controlled. After 25 min, distillation was stopped. By using sulfuric acid (0.05 N), the contents of the receiving flask were titrated to the endpoint, and the TVB-N was determined as follows:TVB-N = (V × N × 100 × 14)/W
where V is the volume of sulfuric acid (mL), N is the normality of sulfuric acid (0.05 N), and W is the weight of the sample in grams.

#### 2.7.2. pH

Twenty-five camel meat samples were homogenized in 10 mL of distilled water, and the pH was determined by using a pH meter (pH ORION, model 330, Court Vernon Hills, USA), with measurements performed in triplicate.

#### 2.7.3. Color

The CIE L*a*b* color scales were used to measure the color of camel meat samples and were dependent on the opponent color theory, which assumes that human eye receptors perceive color as pairs of opposites. Color was evaluated using a colorimeter (CR-300 spectrophotometer, Konica Minolta Inc., Tokyo, Japan) and calibrated using black and white reference tiles. L*, a*, and b* values were recorded at three different locations on each camel meat sample, and each sample was analyzed in triplicate.

#### 2.7.4. Sensory Panel Evaluations

Small cubic camel meat samples were cooked in a microwave oven (Sanyo, Model EM-G1299V, Jiangsu, China) at 1000 watts (temperature of −90 °C) for 10 min. Panels of 10 experienced food scientists were selected to evaluate the sensory attributes of the cooked camel meat. Scales ranging from 0 to 9 were used to estimate the odor and taste acceptability of camel meat samples. The panelists evaluated the products as acceptable or unacceptable in their taste, flavor, and odor. A nine-point hedonic scale was used: 9 = extremely like, 5 = moderately like, and 1 = extremely dislike.

### 2.8. Statistical Analysis

Data are reported as the mean ± standard deviation (M ± SD). Analysis of variances was used to determine the difference between the groups and storage periods in a completely randomized factorial design [26]. When a significant main effect was detected, the means were separated using Duncan’s multiple range test. Differences between groups with *p* < 0.05 were considered significant.

## 3. Results and Discussion

### 3.1. Activity of Citrox and Chitosan toward C. jejuni

The administration of 100 µL of a 1% or 2% Citrox solution evidently inhibited the growth of *C. jejuni*. However, the zone of inhibition was larger for 2% Citrox than for 1% Citrox. However, when using 1% chitosan mixed with 1% or 2% Citrox, there was a more effective and larger zone of inhibition against *C. jejuni* than when using Citrox alone (Figure 1). Therefore, we applied treatments of 1% or 2% Citrox with or without 1% chitosan to camel meat samples and compared the abilities of the treatments using 1% or 2% Citrox alone to stop or control the growth of *C. jejuni* in meat samples during storage.

### 3.2. Total Viable Count

The TVC is a microbiological parameter used to determine the counts of *C. jejuni* in camel meat samples. Figure 2 shows the total viable count (TVC) of *C. jejuni* isolated from camel meat sample by pour plate method and using serial dilution of 10-10^3^/g. The colonies appeared as greyish to white on *Campylobacter* blood-free selective medium (modified CCDA-Preston) after incubation at 42 °C and under a microaerophilic atmosphere in the presence of CO_2_. 

The TVCs of *C. jejuni* in camel meat during 30 days of storage at 10 °C are presented in Figure 3. The camel meat samples treated with *C. jejuni* solution (Group B, positive control) exhibited the highest TVC of ~4.5 log cfu/g after 12 days of storage (Figure 3). Then, the TVC gradually decreased with continuous growth until it reached 1.5 log cfu/g at the end of the storage period of 30 days (Figure 3). However, the TVC in the camel meat samples treated with 0.85% NaCl (Group A, negative control) decreased gradually from 3.5 log cfu/g at the beginning of the storage period to <100 cfu/g by the 18th day of storage (Figure 3). Group C (1% Citrox) camel meat samples stored at 4 °C displayed a gradual decrease in TVC from 3.5 log cfu/g to <100 cfu/g by the 17th day of storage, while the TVC in Group D (2% Citrox) quickly decreased to <100 cfu/g after the 9th day of storage (Figure 3).

The best treatment for decreasing the TVC of *C. jejuni* was the treatment applied to Group F, in which the TVC decreased rapidly from log 3.5 cfu/g to <100 cfu/g by the 6th day of storage. Additionally, for Group F, the TVC of *C. jejuni* decreased by the 10th day of storage at 10 °C (Figure 3).

The TVCs of *C. jejuni* in camel meat during 30 days of storage at 4 °C are presented in Figure 4. The camel meat samples treated with the *C. jejuni* inoculum (Group B, positive control) recorded the highest TVC of 3.8 log cfu/g; then, the TVC gradually reduced with continuous growth until it reached 2.0 log cfu/g after 30 days of storage (Figure 4). However, in the camel meat samples treated with 0.85% NaCl (Group A, negative control), the TVC decreased gradually from 3.5 log cfu/g to <100 cfu/g on the 18th day of storage. The Group C (treated with Citrox 1%) samples stored at 4 °C revealed a gradual decrease in the TVC that was slower than that for Group A, with the TVC decreasing from 3.5 to <100 cfu/g by the 18th day of storage, while the TVC in Group G (Citrox 2%) decreased quickly and reached <100 cfu/g after 17 days of storage.

The best treatment for camel meat stored at 4 °C consisted of 2% or 1% Citrox (Groups E and D) mixed with 1% chitosan. These treatments controlled the growth and TVC of *C. jejuni*, and the growth rate decreased quickly after three and six days. This finding indicates that the antimicrobial (anti-*C. jejuni*) activity of Citrox and chitosan together was higher than that of Citrox alone, with the *C. jejuni* growth curve entering the decline phase very quickly with the former treatment. Camel meat samples stored at 4 °C exhibited no contamination by *C. jejuni,* unlike those stored at 10 °C. Storage temperature is considered one of the most important factors affecting microbial growth [27,28,29]. A storage temperature of 4 °C allows *Campylobacter* to remain viable in a food product during storage [30]. Many studies have mentioned that *Campylobacter* cells do not grow but do survive at 4 °C. This inference is based on a lack of viability being observed when cells are cultured on agar medium, but signs of physiologic and metabolic activity, such as oxygen consumption, catalase activity, Adenosine triphosphate (ATP) generation, chemotaxis, aerotaxis, and protein synthesis, are observed [30]. However, the TVCs increased in our study and in a study conducted by Chan et al. (2001) [31]. A possible explanation comes from Hazeleger et al. (1998) [30], who observed variations in the phenotypes and genotypes of *Campylobacter* species and a variation in survival between different strains. Therefore, it is possible that at 4 °C, only a few strains can grow, while the majority of strains can only survive.

Microbial communities change depending on the temperature [29,32]. According to European Food Safety Authority (EFSA) estimates, microbial communities can be considerably reduced if all slaughtered poultry batches comply with microbiological criteria, with a critical limit of 1000 or 500 cfu/g in neck and breast skin [33]. To reliably quantify the extent of *C. jejuni* contamination in such samples, appropriate (rapid, accurate, reliable, and reasonably priced) enumeration methods should be used. The storage of packaged camel meat samples under vacuum at 4 or 10 °C is important because *Campylobacter* is microaerophilic, making such conditions necessary to reduce its growth and survival. Microaerophilic conditions help *Campylobacter* cope with oxidative stress and toxic products, which are produced during oxygen metabolism. Therefore, bacteria can survive in foods in numbers sufficient to cause infection, despite the constraints imposed by this sensitivity [34]. Aerotolerance has also been reported in numerous studies (Vercellone et al., 1990) [35], and it has even been suggested that *Campylobacter* spp. can adapt to aerobic metabolism [36].

### 3.3. pH

During the storage of camel meat samples at 10 °C, reductions in the pH were noted for the six groups (Figure 5). The pH of the camel meat samples treated with NaCl (saline solution) was 6.0, and this value was reduced to approximately 4.0 after 30 days of storage. Camel meat samples treated with 1 and 2% Citrox combined with 1% chitosan had an initial pH of 5.0 to 5.5, which reached ~4.0 after 30 days of storage. Highly perishable food, such as meat, which is stored under vacuum, shows a reduction in pH due to an increase in the content of microorganisms that cause spoilage. The relative decrease in the pH of the salt-treated camel meat samples compared with that of the *C. jejuni*-contaminated camel meat samples could be attributed to the ionization of NaCl. Camel meat treated with 1 or 2% Citrox also exhibited a reduction in pH, but this reduction was more than that observed for the samples treated with saline solution and *C. jejuni* due to the natural components of the Citrox solution, which include different acids.

Throughout the storage of the camel meat samples at 4 °C, there was a reduction in pH in the six groups (Figure 6). The pH of the camel meat samples treated with 0.85% saline solution was 6.0, and this value was reduced to approximately 5.0 after 30 days of storage. The pH of the camel meat sample treated with *C. jejuni* and/or 0.85% salt was reduced to 5.0. However, the camel meat samples treated with Citrox at 1% or 2% and 1% chitosan with an initial pH ranging from 5.0 to 5.5 reached a pH of ~3.0 after 30 days of storage. The substantial decrease in the pH of the camel meat samples treated with 1% or 2% Citrox and 1% chitosan in comparison with that of the camel meat samples treated with *C. jejuni* or salt during storage at 4 °C could be attributed to the acid components of Citrox and a lack of active microorganisms, which could lead to changes in pH. *Campylobacter* spp. are sensitive to strong acids, such as formic, acetic, ascorbic, and lactic acids, as shown in several studies [37].

The proteolytic activity of *C. jejuni* accelerates the reduction in pH, but this reduction is limited by a low temperature (4 °C) during storage. Rio et al. (2007) [38] reported that dipping chicken meat in citric acid significantly decreased its pH after marination. The pH of meat products is influenced by many factors during storage, such as the duration of storage, the stability of the water-binding capacity and texture, and the redness [39]. *C. jejuni* grew well at an optimum pH range of 6.5 to 7.5, while all strains grew well at a pH range of 5.5 to 8.0 [40]. At pH 5.0, no *C. jejuni* survivors were detected after 24 h, and at pH 9.0, the counts decreased rapidly, with no survivors detected after three days [41].

### 3.4. Total Volatile Base Nitrogen (TVB-N)

The shelf life of meat samples can be determined from the TVB-N released from microbial amino acid decarboxylase activity. In this study, after 30 days of storage at 10 °C, we found that the TVB-N values in camel meat treated with *C. jejuni* alone reached the highest value of 18.5 mg/100 g at 10 °C (Figure 7). This was significantly higher than the TVB-N values of camel meat samples treated with salt (9.95 mg/100 g), 1% Citrox (10.99 mg/100 g), or 2% Citrox (8.9 mg/100 g). However, the samples treated with 1% or 2% Citrox and 1% chitosan were stable, and their TVB-N values did not exceed 8.0 mg/100 g.

After 30 days of storage at 4 °C, the TVB-N values of camel meat treated with *C. jejuni* alone reached their highest value of 14 mg/100 g (Figure 8). This was significantly higher than the TVB-N values of camel meat samples treated with salt (9.88 mg/100 g), 1% Citrox (11.55 mg/100 g), or 2% Citrox (8.66 mg/100 g). However, samples treated with 1% or 2% Citrox and 1% chitosan were stable, and their TVB-N values did not exceed 7.66 mg/100 g. The storage of camel meat samples treated with Citrox and chitosan at 4 °C under vacuum was more effective against *C. jejuni* than storage at 10 °C: storage at 4 °C resulted in higher camel meat quality and improved safety.

TVB-N is an indicator of meat quality and safety. Measuring TVB-N can help determine spoilage in meat. Bell and Garout (1994) [42] explained that there is no consistent relationship between the TVB-N content of the surface and deep tissue of lean beef. They found that the TVB-N content of the surface was 0.2 ± 0.5 mg N/100 g higher than that of the deep tissue, indicating that microflora reached maximal levels (10^7^ cells/cm^2^) in the surface. Moreover, the TVB-N levels exceeded 18 mg N/100 g in lean beef, with levels exceeding 24 mg N/100 g when spoilage became organoleptically evident.

The preservation of meat and meat products using chemical methods to avoid microbial growth is not preferred by consumers. Generally, consumers worldwide dislike the addition of chemical additives to foods. Therefore, demand for the use of natural products as food additives and preservatives has increased. Citrox has plant origins, and in the presence of organic matter, its ingredients, including citric, ascorbic, and malic acids, have the ability to break down bacterial biofilms, extend shelf life, and control foodborne pathogens, such as methicillin-resistant *S. aureus* (Yehia, et al., 2019) [43] and *Listeria monocytogenes* (Tsiraki et al., 2017) [44], as determined by the European suspension test (BS EN 1276). TVB-N has been used in many studies to determine the quantity of biogenic amines produced through the microbiological contamination of foods [45,46,47].

A study on the effect of a 5 min treatment with 1% lactic acid on the counts of *C. jejuni* in broth showed that the TVCs were reduced by 2.1 log units after exposure to low temperature [48], while dipping broiler chicken meat in 0.50% acetic acid or lactic acid for 10 min at 5 °C had little effect, with only a 0.07- and 0.08-log decrease in the counts of *C. jejuni*, respectively [48]. Thus, the effects of organic acids on *Campylobacter* may differ depending on whether testing is performed in broth or in a food matrix.

### 3.5. Colour Values

The six camel meat treatment groups in the present study are shown as color differentials in Table 1. The L* values were significantly higher for the C and D groups (treated with 1% or 2% Citrox solution) than for the positive (B) and negative (A) controls on the 9th day of storage. The L* values of the positive and negative controls were not significantly different. After nine days of storage, the a* values were significantly higher in the samples treated with *C. jejuni* (group B, positive control) than in the samples in the other three groups.

A reduction in the red color of camel meat samples generally indicates the oxidation of lipids and pigments. Citric acid has been used to bind heme iron in myoglobin and prevent the formation of the pink color caused by acidification [49].

We found that the organic acid preservatives present in Citrox solution, such as citric, malic, and ascorbic acids, increased the lightness and redness and decreased the yellowness of the meat. These results are in agreement with those of Bilgili et al. [50], who reported that propionic acid had a small effect on the lightness and redness but significantly decreased the yellowness. The processing conditions during sample preparation include many intrinsic and extrinsic factors, such as pH, temperature, storage time, and immersion chilling conditions, which also change the color of meat [51,52]. Additionally, the use of lactic acid at low concentrations affects the color of fresh and cooked meat [53]. Lactic acid at concentrations of 1.2% and 1.5% caused color deterioration in beef samples during display [53]. Additionally, Kotula and Thelappurate [54] showed that the addition of lactic acid to meat is important when determining the color of fresh and cooked meat.

The formation of metmyoglobin is considered as one of the factors affecting the color of specially packaged fresh beef. The reduction in metmyoglobin in fresh beef requires the removal or injection of oxygen at saturation levels in environmental packaging [55]. However, Djamel et al. [56] suggested that preservation of the red pigment in meat in its oxygenated form (MbO2) requires the use of a high-O_2_ atmosphere. They also suggested that using O_2_ at high levels can promote lipid oxidation, leading to MetMb accumulation.

Using Citrox solution, which contains citric, malic, and ascorbic acids, decreases the redness and increases the lightness of meat [43]. Additionally, Bilgili et al. [50] reported that using propionic acid had little effect on lightness and redness but significantly decreased yellowness.

The conditions used during slaughtering, such as the scalding temperature, storage time [51,52,57], pH (Heath and Wabeck, and immersion chilling conditions (Lyon and Cason, [53], affect the color of meat. Kim et al. [58] reported that the immersion of meat in citric acid solution prevented the increase in redness in meat products conferred by the sous-vide process. An increase in citric acid concentration had reduced the pink color by inducing the thermal denaturation of myoglobin during refrigerated storage.

### 3.6. Sensory Evaluation

A sensory evaluation of the camel meat quality indicated that the samples treated with *C. jejuni* (group B) had a lower sensory score than the samples in the negative control (A) and the other four groups (groups C, D, E, and F; Table 2). Sensory tests performed with Groups C and D (treated with 1% and 2% Citrox, respectively) revealed moderate sensory scores for color, odor, flavor, taste, and tenderness. The use of 1% or 2% Citrox mixed with 1% chitosan resulted in higher scores for color, odor, flavor, taste, tenderness, and overall acceptability than any other treatment.

The addition of citrus extract (0.1 mL/100 g) alone and in combination with an oxygen absorber reduced the TVCs in aerobically packaged ground chicken meat by 0.5 and 1.5 log cfu/g, respectively [59]. Vardaka et al. [60] combined citric extract and chitosan to treat turkey meat to improve its taste and odor, and their results were in agreement with the findings of Petrou et al. [61], who reported that chitosan, applied either alone or in combination with oregano, did not negatively influence the taste of chicken breast meat. Both chitosan and Citrox were sensorially acceptable when added to turkey samples, with chitosan characterized as having spicy, fruity, and oriental flavors and Citrox characterized as having a citrus-like flavor. Moreover, the addition of both citrus extract and chitosan may provide new flavors and options for poultry products. However, further sensory tests are needed to examine this possibility [60].

In another study, a turkey sample sensory profile was acceptable after treatment with chitosan and Citrox. The addition of chitosan improved the flavor, with unique spicy and fruity odors, while Citrox was characterized by citric-like flavors. Moreover, a combination of Citrox and chitosan imparted a new flavor and taste to poultry, but more sensory analysis is still needed to evaluate all of these possibilities [60].

The beneficial effect of chitosan on the quality and sensory profile of meat was attributed to its antibacterial activity and ability to prevent lipid oxidation and preserve color and nutrients, all of which persist in fresh products [62].

Lawrie [63] mentioned that the change in fresh or frozen beef flavor during storage was caused by the slow loss of highly volatile substances. However, changes in odor or taste during beef storage were due to the growth of microorganisms and the breakdown of their chemical components. Some microorganisms secrete the enzyme lipase and split lipids into fatty acids, with more or less unpleasant consequences according to their nature. The off-odor of beef depends on the nature of the beef (fresh, cured, or comminuted) and the conditions, such as the storage temperature, that the microorganism is growing in. It is generally accepted that the autoxidation of membrane phospholipids is largely responsible for the development of an off-odor [64]. However, Calkins and Hodgen [65] stated that high polyunsaturated fatty acid contents in diets may contribute to the appearance of off-flavors in fresh beef muscles.

The addition of Citrox and chitosan to turkey meat led to an antibacterial effect against the growth of *E. coli* and *S. enterica*, which was attributed to the disintegration of the protective outer membrane by Citrox, which could increase the sensitivity of the cell to chitosan. The nature of the chitosan polymer and its emulsification stabilized and enhanced the efficiency of citrus essential oils when used synergistically [60].

## 4. Conclusions

The shelf life of fresh camel meat is highly dependent on many intrinsic and extrinsic factors, such as the storage temperature, the pH, water activity, microbial contamination by pathogens, lipid oxidation, and color changes, and if these factors are controlled, the shelf life can be extended. One of the ways to extend the shelf life of fresh beef is by using organic acids, such as Citrox, alone or in combination with chitosan. Citrox mixed with chitosan controlled the growth of *C. jejuni* at 4 and 10 °C, and the results of this study indicate that Citrox plus chitosan could be used as an antimicrobial treatment against *Campylobacter jejuni* growth in vacuum-packed camel meat stored under refrigerated conditions at 4 °C or at a mild temperature of 10 °C, where populations of this pathogen were maintained at low levels.

## Figures and Tables

**Figure 1 animals-11-01152-f001:**
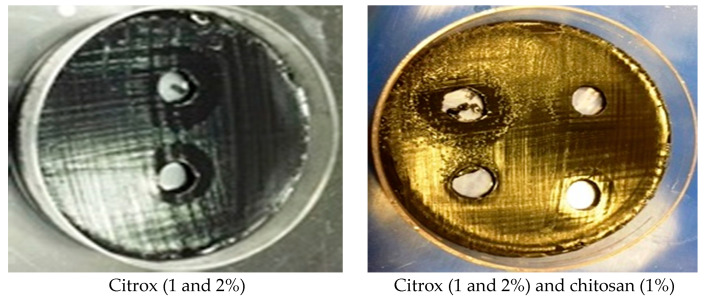
Effect of citrox solution at a concentrations of 1% and 2% citrox and 1% chitosan on *Campylobacter jejuni.*

**Figure 2 animals-11-01152-f002:**
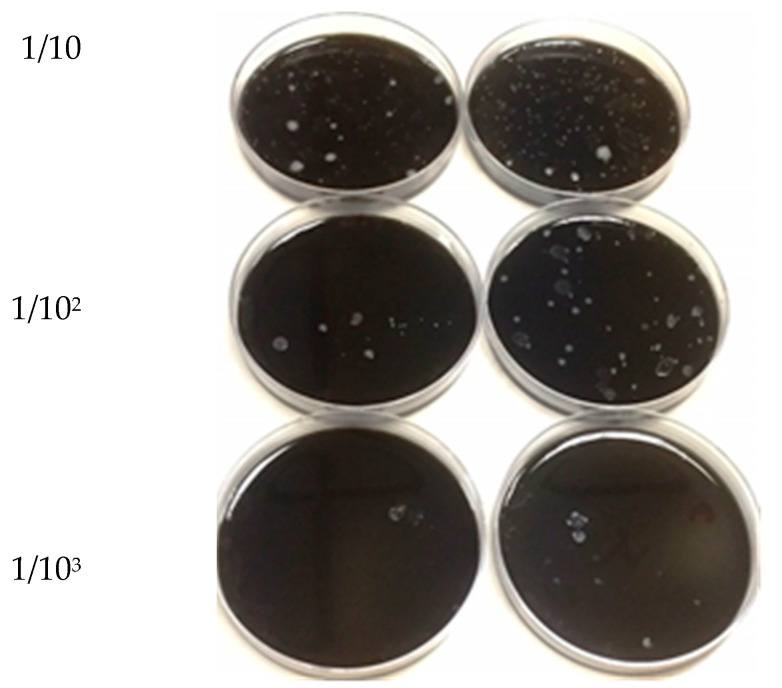
Total viable counts (TVCs) of *Campylobacter jejuni* ATCC 33291 isolated from camel meat after treatment and cultured on *Campylobacter* blood-free selective medium (modified CCDA-Preston).

**Figure 3 animals-11-01152-f003:**
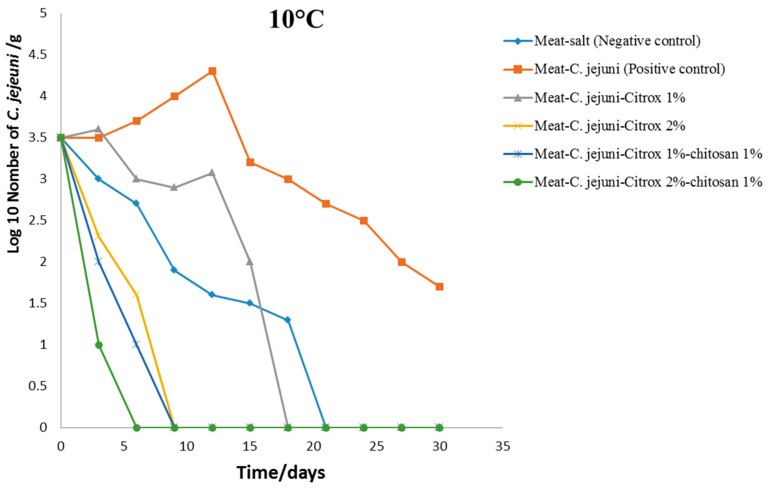
Total viable counts (TVCs) of *Campylobacter jejuni* in camel meat samples stored at 10 °C.

**Figure 4 animals-11-01152-f004:**
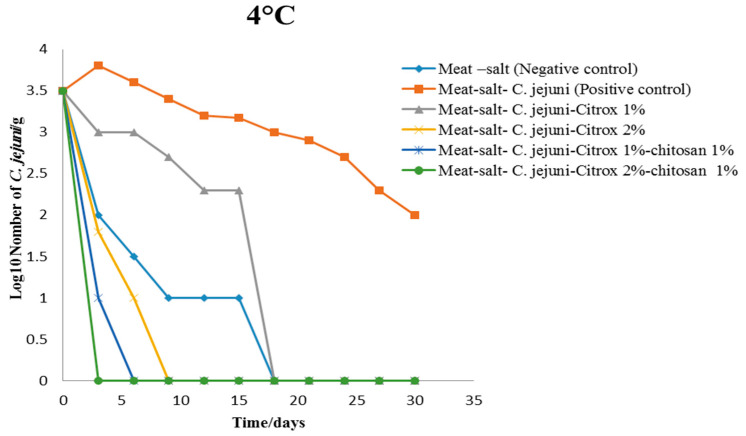
Total viable counts (TVCs) of *C. jejuni* in camel meat samples stored at 4 °C.

**Figure 5 animals-11-01152-f005:**
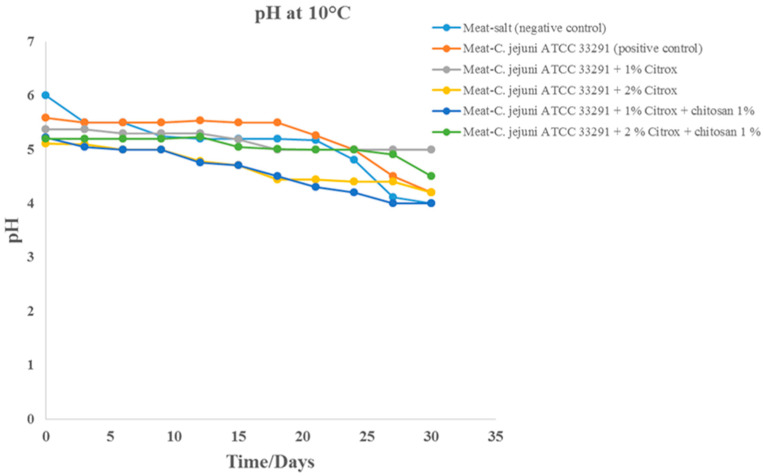
pH values of camel meat samples during storage at 10 °C for 30 days.

**Figure 6 animals-11-01152-f006:**
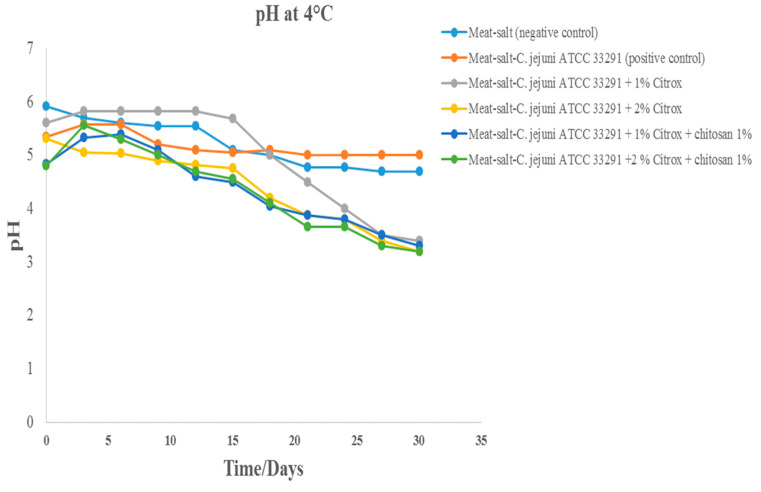
pH values of camel meat stored at 4 °C for 30 days.

**Figure 7 animals-11-01152-f007:**
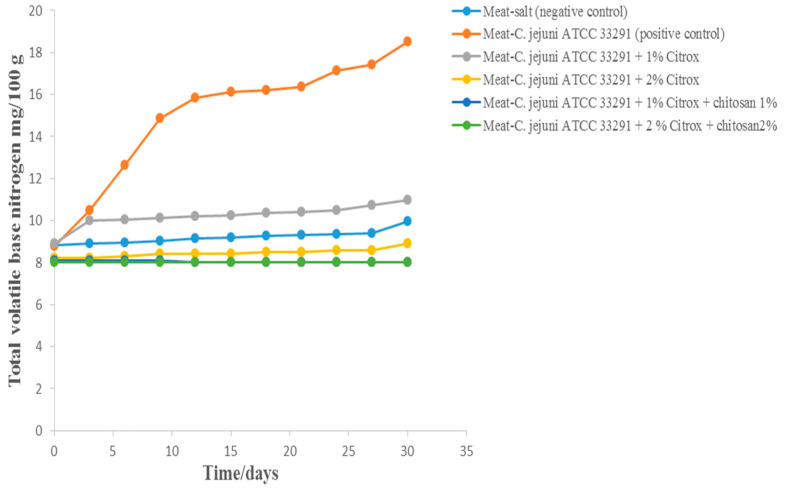
Total volatile base nitrogen (TVB-N) of camel meat samples during storage at 10 °C for 30 days.

**Figure 8 animals-11-01152-f008:**
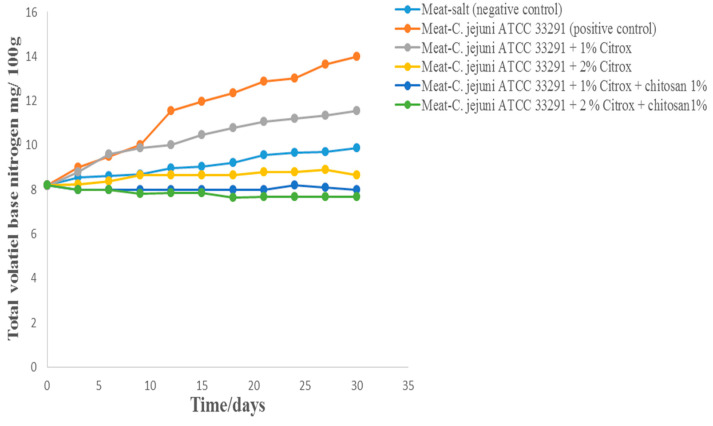
Total volatile base nitrogen (TVB-N) of camel meat samples during storage at 4 °C for 30 days.

**Table 1 animals-11-01152-t001:** Hunter color values of camel meat samples treated with Citrox and chitosan and stored at 4 or 10 °C for 30 days.

Hunter Color			0		3		6		9		12		15		18		21		24		27		30		
		Temperatures (°C)	4	10	4	10	4	10	4	10	4	10	4	10	4	10	4	10	4	10	4	10	4	10	Mean
	Treatments																								
L *	M-S (A)		39.76	38.25	40.67	38.47	40.81	38.91	40.45	40.23	40.51	40.39	47.93	51.6	46.47	48.29	45.74	47.13	45	45.05	44.99	42.2	44	43	43.17 ^e^
	M-S-*C. jejuni* (B)		37.71	37.93	43.5	41.33	43.18	41.48	45.07	42.02	45.22	44.48	45.65	45.72	46.07	48.93	43.88	46.21	43.84	4487	41.2	42.84	41	42.02	43.37 ^c^
	M-S-*C. jejuni*-C 1% (C)		39.33	38.37	40.24	39.11	44.22	39.39	49.1	43.17	49.45	44.79	49.21	45.22	48.81	46.7	45.07	45.24	43.09	43.01	42.41	40.96	42.01	40.54	43.36 ^d^
	M-S-*C. jejuni*-C 2% (D)		34.92	38.86	39.21	43.55	40.29	43.96	42.65	44.2	44.46	47.7	47.93	46.68	47.19	46.33	46.63	46.14	44.45	45.67	42.6	42.77	42	42.66	43.61 ^b^
	M-S-*C. jejuni*-C 1%–C 1% (E)		41.87	39.64	42.15	42.8	44.78	48.41	45.43	50.98	47.17	50.92	48.93	53.58	48.2	49.63	46.63	49.94	44.83	48.2	42.5	46.38	42.52	45	46.38 ^a^
	M-S-*C. jejuni*-C 1%–C 2% (F)		40.3	40.96	42.77	40.99	43.21	41.54	47.18	44.46	47.53	46.6	46.36	48.73	45.5	44.1	45.4	44	44.56	43.52	43.2	43	42.33	42.36	44.02 ^f^
	Mean		38.98	39	41.42	41	42.74	42.28	44.98	44.17	45.72	45.81	47.66	48.58	47.04	47.33	45.55	46.44	44.29	45.05	42.81	43.02	42.31	42.59	-
a *	M-S (A)		6.7	10.35	8.69	12.23	7.7	9.9	7.27	9.42	10.97	9.38	12.1	8.82	9.47	7.48	9.05	6.92	6.65	6	6.5	5.46	6	5.44	8.29 ^e^
	M-S-*C. jejuni* (B)		10.46	8.35	11.9	8.52	12.4	12.5	13.8	12.49	12.8	12.67	11.62	11.49	11.44	10.99	11.34	8.68	8.03	7.05	6.48	6.25	6.25	7.58	10.14 ^b^
	M-S-*C. jejuni*-C 1% (C)		6.07	9.65	9.06	9.34	9.78	9.5	11.7	10.84	12.57	12.83	14.21	9.92	11.45	8.78	8.06	6.81	7.3	7.81	5.16	6.58	6.5	6	9.08 ^c^
	M-S-*C. jejuni*-C 2% (D)		6.82	4.51	7.77	9.1	8.31	9.19	8.81	10.42	15.97	11.79	15.38	13.59	13.48	10.95	12.79	10.49	11.97	9.76	8.92	9.59	6	8	10.16 ^a^
	M-S-*C. jejuni*-C 1%–C 1% (E)		6.95	9.57	8.35	9.89	8.24	10.65	9.18	11.06	11.71	8.85	11.15	8.25	9.79	7.99	9.09	7	8.25	7	6.05	6.2	6	4.47	8.44 ^d^
	M-S-*C. jejuni*-C 2%–Ch 1% (F)		6.25	3.53	6.19	5.45	6.61	6.41	12.32	7.94	12.94	7.43	8.21	8.57	7.12	8.97	7.1	7	5.08	6	6	6	5.99	4.56	7.07 ^f^
	Mean		7.2	7.66	8.66	9.08	8.84	9.69	10.51	10.36	12.82	10.49	12.11	10.1	10.45	9.19	9.57	7.81	7.88	7.27	6.51	6.68	6.12	6	-
b *	M-S (A)		4.27	6.29	4.58	6.23	5.88	7	7.42	7.5	7.65	7.99	8.79	8	9.98	8.95	10.24	9.86	9.89	7	8.02	6.66	6	5.99	7.46 ^e^
	M-S-*C. jejuni* (B)		7.21	6.22	7.83	6.65	8.32	7.37	8.5	8.05	9.8	7.58	11.55	9.69	8.02	8.99	6.4	6.58	5.39	6.43	5.94	4.59	5.5	5	7.34 ^f^
	M-S-*C. jejuni* -C 1% (C)		6.51	5.72	7.04	6.33	8.12	7	9.11	7.5	9.92	8.7	10.77	8.7	10.28	8.78	7.63	7.2	6.78	6.3	6.48	5.56	6	5.12	7.52 ^d^
	M-S-*C. jejuni*-C 2% (D)		6.96	5.62	7.7	7.03	8.84	9.45	11.73	9.53	11.07	10.35	10.92	11.92	8.45	11.11	5.66	9.75	5.38	9.99	4.64	8.19	4.56	4.38	8.32 ^c^
	M-S-*C. jejuni*-C 1%–C 1% (E)		6.6	6.99	7.74	9.41	8.03	9.88	9.02	10.19	9.92	0.241	12.23	10.96	10.6	11.26	9.71	9.21	9.23	8.83	6.23	8.63	6	5.66	8.48 ^b^
	M-S-*C. jejuni*-C 2%–Ch 1% (F)		6	7.14	8.47	10.92	9.72	9.66	9.85	11.28	11.43	12.95	11.17	11.63	10.63	10.49	9.53	10	9	8.55	8.5	8	6	6	9.40 ^a^
	Mean		6.25	6.33	7.227	7.76	8.157	8.39	9.27	9	9.965	7.968	10.9	10.15	10.15	9.93	8.195	8.767	7.61	7.85	6.63	6.93	5.67	5.35	-

• L * scale: Light vs. dark, where a low number (0–50) indicates dark and a high number (51–100) indicates light. • a * scale: red vs. green, where a positive number indicates red and a negative number indicates green. • b * scale: yellow vs. blue, where a positive number indicates yellow and a negative number indicates blue. • M-S (group A) = meat-salt, M-S-*C. jejuni* (group B) = meat-salt-*C. jejuni,* M-S-*C. jejuni*-C 1% (group C) = meat-salt-*C. jejuni +* Citrox 1%, M-S-*C. jejuni*-C 2% (group D)= meat-salt *C. jejuni +* Citrox 2%, M-S-*C. jejuni*-C1%-Ch 1% (group E)= meat-salt *C. jejuni +* Citrox 1% + chitosan 1%, M-S-*C. jejuni*-C 1%- Ch 1% (group F) = meat-salt-*C. jejuni* + Citrox 2% + chitosan 1%. ^a, b, c, d, e, f^ Different superscripted letters indicate significant differences (*P ≤ 0.05*) among the observed values within columns.

**Table 2 animals-11-01152-t002:** Sensory evaluation of camel meat samples treated with 1% or 2% Citrox.

Treatments	Color	Odor	Flavor	Taste	Tenderness	Overall Acceptability
Camel meat-salt (group A)	5.0 ^c^	4.0 ^d,e^	4.5 ^c^	5.0 ^c^	4.5 ^b^	5.2 ^c^
Camel meat-*C. jejuni* (group B)	3.0 ^d^	2.0 ^f^	2.0 ^d^	2.0 ^e^	2.3 ^d^	2.5 ^e^
Camel meat-*C. jejuni*- Citrox 1% (group C)	4.9 ^a,b,c^	4.65 ^d^	4.6 ^c^	4.9 ^d^	4.5 ^b^	4.0 ^d^
Camel meat-*C. jejuni*- Citrox 2% (group D)	5.0 ^c^	5.0 ^c^	4.9 ^a,b^	5.0 ^c^	4.4 ^b^	4.0 ^d^
Camel meat-*C. jejuni*- Citrox 1%- Chitosan 1% (group E)	5.5 ^a,b^	6.0 ^a,b^	5.0 ^b^	5.5 ^b^	4.5 ^b^	6.0 ^b^
Camel meat-*C. jejuni*- Citrox 2%- Chitosan 1% (group F)	6.0 ^a^	6.2 ^a^	5.5 ^a^	6.0 ^a^	4.6 ^a^	6.4 ^a^
Mean	4.9 ^a^	4.6 ^c,d^	4.41 ^e^	4.73 ^b^	4.0 ^f^	4.68 ^c^

A nine-point hedonic scale (9 = like extremely, 4–5 = like moderately, 1 = dislike extremely) was used for all sensory parameters. ^a, b, c, d, e, f^ Different superscripted letters indicate significant differences (*P ≤ 0.05*) among the observed values within columns.

## Data Availability

The raw data of the results presented in this study are available on request from the corresponding author.

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
