# Peer review of "Improving the Quality and Safety of Fresh Camel Meat Contaminated with Campylobacter jejuni Using Citrox, Chitosan, and Vacuum Packaging to Extend Shelf Life"

_animals, 2021, doi:10.3390/ani11041152_

Round 1
Reviewer 1 Report
Manuscript entitled: Improving the quality and safety of fresh camel meat contaminated by Campylobacter jejuni using citrox and chitosan and vacuum packaging to extend shelf life; describes the study of using an antimicrobial strategy to reduce Campylobacter and preserve quality of camel meat. The manuscript is interesting; however, it suffers from numerous and serious limitations.
- Abstract must be improved to depict more precisely the results and significance of the present study. The authors should avoid the use of ambiguous phrases regarding the impact of the results.
- The authors should revise and reconsider interpretation of results. In food microbiology, it is not possible to assume that a product possesses or reach 0 log cfu/g. This idea should be revised.
- Citrox seems to be the trade name of a commercial product. The authors should revise and clarify.
- The number of samples used for the analysis seems odd. It describes that 792 meat samples were used, but then it is mentioned that each group, A, B, C, D, E, and F, contained 264 samples. The authors should clarify this concern.
- Overall, the whole experimental design suffers of fundamental flaws. After inoculation with the pathogen, the samples were soaked in a solution. Thus, it is very possible that the observed effects are due to the removal of bacteria by this process.
- There is no evidence of experimental replication and the depicted data lacks dispersion measurements; as well as statistical assessment.
- In general, results are poorly portrayed, for example figures 1 and 2. Overall, and as presented, it is exceedingly difficult to appreciate the robustness and significance of the present study.
Author Response
Dear Editor of Animal Journals
Thank you very much about your advices about our manuscript no., animals-1135298
Title:
Improving the quality and safety of fresh camel meat contaminated by Campylobacter jejuni using citrox and chitosan and vacuum packaging to extend shelf life
And the following is the replying for every Reviewer point by point
Reviewer 1
English language and style
( ) Extensive editing of English language and style required
(x) Moderate English changes required
( ) English language and style are fine/minor spell check required
( ) I don't feel qualified to judge about the English language and style
English Editing corrected again by Specific office (Springer English Editing)
|
|
Can be improved |
Must be improved |
Not applicable |
|
|
Does the introduction provide sufficient background and include all relevant references? |
( ) |
(x) |
( ) |
( ) |
|
Is the research design appropriate? |
( ) |
( ) |
(x) |
( ) |
|
Are the methods adequately described? |
( ) |
(x) |
( ) |
( ) |
|
Are the results clearly presented? |
( ) |
( ) |
(x) |
( ) |
|
Are the conclusions supported by the results? |
( ) |
( ) |
(x) |
( ) |
Comments and Suggestions for Authors
-The introduction and background were improved
-The methods also improved
- Abstract must be improved to depict more precisely the results and significance of the present study. The authors should avoid the use of ambiguous phrases regarding the impact of the results.
The abstract rewrite again and abbreviated to the most important results
- The authors should revise and reconsider interpretation of results. In food microbiology, it is not possible to assume that a product possesses or reach 0 log cfu/g. This idea should be revised.
Yes right (if there is no growth of bacteria it was written as ˂100 cfu/g) in Line 31 and when mention in all the manuscript and I was changed in all the manuscript
- Citrox seems to be the trade name of a commercial product. The authors should revise and clarify.
Yes, Citrox is trade name of commercial product and knew its composition so we prepared it in the laboratory from pure chemicals and sterilized as many authors used in different researches.
- The number of samples used for the analysis seems odd. It describes that 792 meat samples were used, but then it is mentioned that each group, A, B, C, D, E, and F, contained 264 samples. The authors should clarify this concern.
Yes excuse me each contained 198 samples as follows:;
We used 6 treatments (A, B, C, D, E and F),
And three replicate for each = 18, then every three days one sample (0, 3, 6, 9, 12, 15, 18, 21, 24, 27, 30 days) , so, The total = 198 samples for the total count
And the same for pH= 198 samples
And the same for TVC=198
And sensory test =198
THE TOTAL = 792
So I was correct it to 198 samples in line 120
- Overall, the whole experimental design suffers of fundamental flaws. After inoculation with the pathogen, the samples were soaked in a solution. Thus, it is very possible that the observed effects are due to the removal of bacteria by this process.
Of course the number is decreased so if we used inoculum of 4.0 log10 cfu/g it was reached to 3.5 log10 cfu/g at 0 time, but soaking in citrox and/or chitosan must added as antimicrobial at the end. As sometimes in many researches chitosan used as suitable material for food packaging purposes.
- There is no evidence of experimental replication and the depicted data lacks dispersion measurements; as well as statistical assessment.
Three replicates and we statically used Mean and SD.
- In general, results are poorly portrayed, for example figures 1 and 2. Overall, and as presented, it is exceedingly difficult to appreciate the robustness and significance of the present study.
Dear Professor, I respect your opinion but you may share me that the dealing with microbe as Campylobacter jejuni is very difficult specially the condition of growth from using jar with gas generators, incubation temperatures at 42 °C, complicated media and the package under vacuum condition. So the data is very important specially also that there is no or little works on treatment of C. jejuni by different natural substances as citrox and chitosan on camel meat too.
___________________________________________________________________
Reviewer 2
English language and style
( ) Extensive editing of English language and style required
(x) Moderate English changes required
( ) English language and style are fine/minor spell check required
( ) I don't feel qualified to judge about the English language and style
English Editing Repeated again by Specific office (Springer English Editing)
|
Yes |
Can be improved |
Must be improved |
Not applicable |
|
|
Does the introduction provide sufficient background and include all relevant references? |
( ) |
( ) |
(x) |
( ) |
|
Is the research design appropriate? |
(x) |
( ) |
( ) |
( ) |
|
Are the methods adequately described? |
(x) |
( ) |
( ) |
( ) |
|
Are the results clearly presented? |
( ) |
( ) |
(x) |
( ) |
|
Are the conclusions supported by the results? |
( ) |
( ) |
(x) |
( ) |
The introduction and background were improved
The results explained clearly with the addition of some other paragraphs.
Also conclusions were improved
- the paper is missing a better explanation about what Citrox is. You infer it is a mixture of organic acids but it is not clear
its composition from itrus extracts and organic acids so I was explained in Line 75-82 more details about citrox.
- For the results and discussion the microbiological analysis is fin
We try to improve the results and discussion again by specific English editing Springer, and I was sending the modification copy in supplementary materials.
- The pH analysis is repetitive and there is no mention about pH changes due to aging of meat
The changing in pH values are expected to reduced but not too much because first, the time between every reading of pH not so big (every three days) and second C. jejuni growing slowly with the reduction in TVC (the log phase not continued so much time).
- A subtitle introducing the TVBN results is required. You jump form pH results to TVBN results and there is no subtitle. Line 358 to 360 is very confusing. You are comparing cattle to camel but then you are stating that TVBN values are higher in cattle except for adult camel females. Did you treatments include male Vs Female? this is not on the experimental design and it is very confusing
The subtitle of total volatile base nitrogen was added line 349
I was removed this paragraph because it is not clear as you suggest (Line 361-364)
(In the present study, TVBN values did not show a significant increase during frozen storage time. Cattle meat had higher TVBN values than camel meat, except for adult female camels. Additionally, young animals often had higher TVBN values than adults in the two species.)
- Color value. Lines 384 and 391 refer to S. aureus and not C. jejuni. Is this a typo? because we are talking about two very different microorganisms here. The analysis is very repetitive (compare lines 386-389 and then down below 413-416. Lines 394-398 and lines 427-430. Also look at your citation style. you are following numerate style, however line 416 mixes numerate and author name style
Line 379 refer to C. jejuni but by mistake was written S. aureus
I was change it
Line also f the following paragraph 394- 398 was removed
However, a significant increase in the b* value was detected in samples treated only with S. aureus compared with those of the other three treatments. The effects of chilling and using various concentrations of acidic citrox led to a change in the skin color of camel meat.
- Color table is too crowded. very difficult to read and follow
Colour table yes very difficult to read, I know but I try to explain the main changes between treatments.
- sensory analysis. You need to elaborate more on your results.
I added more results in Line 437-440
And also in Line 465- 460
__________________________________________________________________________
Reviewer 3
|
Yes |
Can be improved |
Must be improved |
Not applicable |
|
|
Does the introduction provide sufficient background and include all relevant references? |
( ) |
(x) |
( ) |
( ) |
|
Is the research design appropriate? |
( ) |
( ) |
(x) |
( ) |
|
Are the methods adequately described? |
( ) |
( ) |
(x) |
( ) |
|
Are the results clearly presented? |
( ) |
( ) |
(x) |
( ) |
|
Are the conclusions supported by the results? |
( ) |
( ) |
(x) |
( ) |
The introduction and background were improved
The research design is improved
The method also improved
The results explained clearly with the addition of some other paragraphs.
Also conclusions were improved
Point 1: You have mentioned “Seven hundred and ninety-two camel meat samples, weighing approximately 25 g each, were divided into six main groups, A, B, C, D, E, and F, which each contained 264.”As per my understanding:
Total sample= 792
6 groups and each contaiing 264 samples = 6 x 264= 1584.. (not 792).
We used 6 treatments (A, B, C, D, E and F),
And three replicate for each = 18, then every three days one sample (0, 3, 6, 9, 11, 14, 17, 20, 23, 26, 29 days) , so, The total = 198 samples for the total count
And the same for pH= 198 samples
And the same for TVC=198
And sensory test =198
So the total = 792
So I was correct it to each 198 samples in line 130
Point 2: 4.1. Treatments groups
All the study design needs a lot of time and attention of reader. Cannot you add a schematic so that reader can understand it quickly and follow it throughout the paper. Suggest adding schematics for this study design.
Do you suggest like this diagram ??????
Group-B: It is positive control, you named it group-B. Why not name it simply control group to ease of reader and then keep other groups A-E + control group.
We try to mention the details for every treatment.
Point 3: 7.4. Sensory panel evaluations
Camel meat samples were cooked using a microwave oven (Sanyo, Model: EM- 200
G1299V, Jiangsu, China) on high power for 10 min
What was the temperature of the over? And at which temperature meat was served to the panel?
In microwave at 1000 watt the temperature was about 90°C and the time 10 min
Point 4: Results and discussion: Can you add more discussion relating it to the results from sensory panel.
I was added more discussion and added new references
Line 437-440
And also in Line 465- 460
Why tenderness in all treatment groups is similar? Any discussion on that would help reader’s
understanding?
Four of the treatments are very closely related as citrox 1%, citrox 2% , chitosan -citrox ( 1% and 1%), chitosan –citrox ( 1%- 2%)
As you noticed citrox was found in all the four treatments
And the fifth treatment with saline solution also appeared a significant effect on bacteria
Point 5: What do you think about effect of weather? Since this study was conducted in KSA and I may assume a hot climate (just my opinion).
Yes it is a factor can effect on the meat quality and contamination by bacteria and also interfere with many other factors as in summer not as in winter. So I think also the data may changes if this put in our mind in future studies.
Point 6: You have used vacuum packaging on all samples? Do you think it has also some sort of effect of the findings of this study? What if this experiment was also performed on meat stored without vacuum packaging?
Without vacuum as you know Dear Professor that the condition not be suitable for the growth of Campylobacter which prefer a microaerophilic condition.
Point 7: In describing the sensory panel, was any interaction effect between all treatments?
There is interfering found but the using of citrox and chitosan closely in many parameters of sensory test than other treatments.
Point 8: In analyzing the sensory panel data, was effect of panel (as independent variable) taken into consideration?
Yes it was taken in our consideration this variable.
Thank you in advance
Hany

Reviewer 2 Report
- the paper is missing a better explanation about what Citrox is. You infer it is a mixture of organic acids but it is not clear
- For the results and discussion the microbiological analysis is fine
- The pH analysis is repetitive and there is no mention about pH changes due to aging of meat
- A subtitle introducing the TVBN results is required. You jump form pH results to TVBN results and there is no subtitle. Line 358 to 360 is very confusing. You are comparing cattle to camel but then you are stating that TVBN values are higher in cattle except for adult camel females. Did you treatments include male Vs Female? this is not on the experimental design and it is very confusing
- Color value. Lines 384 and 391 refer to S. aureus and not C. jejuni. Is this a typo? because we are talking about two very different microorganisms here. The analysis is very repetitive (compare lines 386-389 and then down below 413-416. Lines 394-398 and lines 427-430. Also look at your citation style. you are following numerate style, however line 416 mixes numerate and author name style
- Color table is too crowded. very difficult to read and follow
- sensory analysis. You need to elaborate more on your results.
Author Response

(The authors gave the same response as above.)

Reviewer 3 Report
Point 1: You have mentioned “Seven hundred and ninety-two camel meat samples, weighing approximately 25 g each, were divided into six main groups, A, B, C, D, E, and F, which each contained 264.”As per my understanding:
Total sample= 792
6 groups and each contaiing 264 samples = 6 x 264= 1584.. (not 792).
Point 2: 4.1. Treatments groups
All the study design needs a lot of time and attention of reader. Cannot you add a schematic so that reader can understand it quickly and follow it throughout the paper. Suggest adding schematics for this study design.
Group-B: It is positive control, you named it group-B. Why not name it simply control group to ease of reader and then keep other groups A-E + control group.
Point 3: 7.4. Sensory panel evaluations
Camel meat samples were cooked using a microwave oven (Sanyo, Model: EM- 200
G1299V, Jiangsu, China) on high power for 10 min
What was the temperature of the over? And at which temperature meat was served to the panel?
Was this panel developed the sensory vocabulary before starting the actual test?
Point 4: Results and discussion: Can you add more discussion relating it to the results from sensory panel.
Why tenderness in all treatment groups is similar? Any discussion on that would help reader’s understanding?
Point 5: What do you think about effect of weather? Since this study was conducted in KSA and I may assume a hot climate (just my opinion).
Point 6: You have used vacuum packaging on all samples? Do you think it has also some sort of effect of the findings of this study? What if this experiment was also performed on meat stored without vacuum packaging?
Point 7: In describing the sensory panel, was any interaction effect between all treatments?
Point 8: In analyzing the sensory panel data, was effect of panel (as independent variable) taken into consideration?
Author Response

(The authors gave the same response as above.)

Round 2
Reviewer 3 Report
Authors have tried to improve in this version. However, there is still gaps especially in describing the Materials and Methods section and results.
Authors have conducted good studies but should try to synchronize them and elaborate them well. Authors should add more comprehensive detail for sensory analysis. It was mentioned that:
In microwave at 1000 watt the temperature was about 90°C and the time 10 min
It would have good effect on the meat itself. But, how authors justify this high temperature.
Authors have more focus on describing the chemicals used rather than making effrorts to describe other parts and then combine all together. Otherwise, reader will be confused.
Authors may note that each table and figure should be easy to understand and self-explanatory.
I still belive, with little efforts the quality of this study can be increased.
Author Response
Authors have conducted good studies but should try to synchronize them and elaborate them well. Authors should add more comprehensive detail for sensory analysis. It was mentioned that:
In microwave at 1000 watt the temperature was about 90°C and the time 10 min
It would have good effect on the meat itself. But, how authors justify this high temperature.
Dear Professor the microwave work depends on the power level used and this equal a main Watt for a limit time, for example if we use the high max level as mentioned in the table below, this used to cook meat, poultry, fish, vegetables, minced lamp and boil water (this mean temperature reached to 100 °C but for meat and poultry reached about 90 °C) and the power output 1000 watt, this all mentioned in manual instruction of using microwave as mentioned of the following table:
Please note: For more details about temperature I was used a table for temperatures used inside the microwave with every kind of foods as in PDF file (upload it with the revised manuscript in supplementary materials)
Authors have more focus on describing the chemicals used rather than making effrorts to describe other parts and then combine all together. Otherwise, reader will be confused.
Yes right we try to compare the difference between using of chemicals as different organic acids and nature compounds as citrox and i was doing English Editing two times trying to be accurate using terms related to our studying.
Authors may note that each table and figure should be easy to understand and self-explanatory.
I still belive, with little efforts the quality of this study can be increased.
I tried to be easy explaining the figures more than this but as you know the treatments too much and also the Table used for color some authors use a separately Figure for L, a and b
Dear author I thank you very much about your suggestion and i have to improve and take in mind for the future researches and if you advise me to change the Table of sensory analysis to figures, i can do it without doubt but do you think that it will be a complicated more than the figures put in the manuscript.
